# CLAREL: CLASSIFICATION VIA RETRIEVAL LOSS FOR ZERO-SHOT LEARNING

## ABSTRACT

We address the problem of learning fine-grained cross-modal representations. We propose an instance-based deep metric learning approach in joint visual and textual space. The key novelty of this paper is that it shows that using per-image semantic supervision leads to substantial improvement in zero-shot performance over using class-only supervision. On top of that, we provide a probabilistic justification for a metric rescaling approach that solves a very common problem in the generalized zero-shot learning setting, *i.e.*, classifying test images from unseen classes as one of the classes seen during training. We evaluate our approach on two fine-grained zero-shot learning datasets: CUB and FLOWERS. We find that on the generalized zero-shot classification task CLAREL consistently outperforms the existing approaches on both datasets.

## 1 INTRODUCTION

Deep learning-based approaches have demonstrated superior flexibility and generalization capabilities in information processing on a wide variety of tasks, such as vision, speech and language (LeCun et al., 2015). However, it has been widely realized that the transfer of deep representations to real-world applications is challenging due to the typical reliance on massive hand-labeled datasets. Learning in the low-labeled data regime, especially in the zero-shot (Wang et al., 2019) and the few-shot (Wang & Yao, 2019) setups, have recently received significant attention in the literature. In the problem of zero-shot learning (ZSL), the objective is to recognize categories that have not been seen during the training (Larochelle et al., 2008). This is typically done by relying on anchor embeddings learned in one modality as prototypes and by associating a query embedding from the other modality with the closest prototype. In the generalized ZSL (GZSL) case (Xian et al., 2018c), the objective is more challenging as recognition is performed in the joint space of seen and unseen categories. ZSL, as well as its generalized counterpart, provide a viable framework to learn cross-modal representations that are flexible and adaptive. For example, in this paradigm, the adaptation to a new classification task based on text/image representation space alignment could be as easy as defining/appending/modifying a set of text sentences to define classes of new classifiers. This is an especially relevant problem as machine learning is challenged with the long tail of classes, and the idea of learning from pairs of images and sentences, abundant on the web, looks like a natural solution. Therefore, in this paper we specifically target the fine-grained scenario of paired images and their respective text descriptions. The uniqueness of this scenario is in the fact that the co-occurance of image and text provides a rich source of information. The ways of leveraging this source have not been sufficiently explored in the context of GZSL. Although we focus exclusively on the GZSL recognition setup in this paper, we believe that the research in this direction has potential to enable zero-shot flexibility in a wider array of high-level tasks such as segmentation or conditional image generation (Zhang et al., 2018). The contributions of this work can be characterized under the following two themes.

**Instance-based training loss.** Most prominent zero-shot learning approaches rely heavily on class-level modality alignment (Xian et al., 2018c). We propose a new composite loss function that balances instance-based pairwise image/text retrieval loss and the usual classifier loss. The retrieval loss term does not use class labels. We demonstrate that the class-level information is important, but in the fine-grained text/image pairing scenarios, most of the GZSL accuracy can be extracted from the instance-based retrieval loss. To the best of our knowledge, this type of training has not been used in the GZSL literature. Its impressive performance opens up new promising research directions.

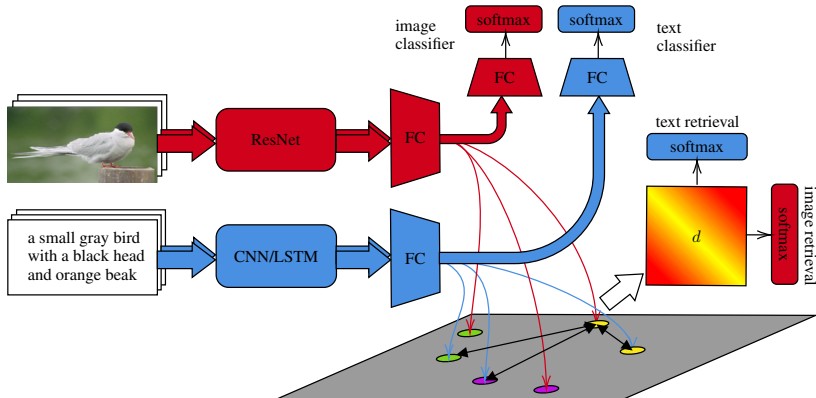

Figure 1: The architecture and training diagram describing the proposed method. Each batch consists of randomly sampled instances, *i.e.* pairs of images and their corresponding texts. Images are embedded via ResNet and texts are embedded via a CNN/LSTM stack. Image and text features are projected via a fully connected layer into the same dimensional space. In this space, distances between text and image features from different instances are computed. The negative distances are fed into softmax to train on both the image and the text retrieval tasks. The image retrieval task consists of retrieving the image corresponding to the given text of the same instance and the text retrieval task is vice versa. In addition to that, image and text embeddings are trained on auxiliary image and text classification tasks on the class labels corresponding to instances.

**Metric space rescaling.** Metric-based ZSL approaches rely on distances between prototypes and query embeddings during inference. They are known to suffer from imbalanced performance on seen and unseen classes (Liu et al., 2018). Previous work proposed to use a heuristic trick, calibrated stacking (Chao et al., 2016) or calibration (Das & Lee, 2019), to solve the problem. We refer to this technique as metric rescaling in our work, and provide a sound probabilistic justification for it.

## 2 PROPOSED METHOD

In this paper, we specifically target the fine-grained visual description scenario, as defined by Reed et al. (2016). In this setting, the dataset consists of a number of images from a given set of classes and each image is accompanied by a number of textual descriptions. The task is to learn a joint representation space for images and texts that can be used for zero-shot recognition. An instance of the zero-shot multimodal representation learning problem can then be defined as follows. Given a training set $\mathcal{S} = \{(v_n, t_n, y_n) \mid v_n \in \mathcal{V}, \ t_n \in \mathcal{T}, y_n \in \mathcal{Y}, n = 1 \ldots N\}$ of image, text and label tuples, we are interested in finding representations $f_\phi : \mathcal{V} \to \mathcal{Z}$ of image, parameterized by $\phi$, and $f_\theta : \mathcal{T} \to \mathcal{Z}$ of text, parameterized by $\theta$, in a common embedding space $\mathcal{Z}$. Furthermore, GZSL problem is defined using the sets of seen $\mathcal{Y}^{tr}$ and unseen $\mathcal{Y}^{ts}$ classes, such that $\mathcal{Y} = \mathcal{Y}^{tr} \cup \mathcal{Y}^{ts}$ and $\mathcal{Y}^{tr} \cap \mathcal{Y}^{ts} = \emptyset$. The training set will then only contain the seen classes, *i.e.* $\mathcal{S}^{tr} = \{(v_n, t_n, y_n) \mid v_n \in \mathcal{V}, \ t_n \in \mathcal{T}, y_n \in \mathcal{Y}^{tr}\}$ and the task is to build a classifier function $g : \mathcal{Z} \times \mathcal{Z} \to \mathcal{Y}$. This is different from the ZSL scenario focusing on $g : \mathcal{Z} \times \mathcal{Z} \to \mathcal{Y}^{ts}$.

To build $g$, most approaches to joint representation learning rely on class labeling to train a representation. For example, all the methods reviewed by Xian et al. (2018c) require the access to class labels at train time. We hypothesise that in the fine-grained learning scenario, such as the one described by Reed et al. (2016), a lot of information can be extracted simply from pairwise image/text co-occurrences. The class labels really only become critically necessary when we define class prototypes, *i.e.* at zero-shot test time. Following this intuition, we define a composite loss function that relies both on the pairwise relationships and on the class labels. The high-level description of the proposed framework is depicted in Figure 1. The framework enables us, among other things, to experiment with the effects of train-time availability of class labels on the quality of zero-shot representations. The framework is based on projecting texts and images into a common space and then learning a representation based on a mixture of four loss functions: a pairwise text retrieval loss, a pairwise image retrieval loss, a text classifier loss and an image classifier loss (see Algorithm 1).

**Algorithm 1** Loss calculation for a single optimization iteration of the proposed method. $N$ is the number of instances in the training set $\mathcal{S}^{tr}$, $B$ is the number of instances per batch, $C$ is the number of classes in the train set. RANDOMSAMPLE$(\mathcal{S}, B)$ denotes a set of $B$ elements chosen uniformly at random from a set $\mathcal{S}$, without replacement.

---

**Input:** Training set $\mathcal{S}^{tr} = \{(v_1, t_1, y_1), \ldots, (v_N, t_N, y_N)\}$, $\lambda \in [0, 1]$, $\kappa \in [0, 1]$.
**Output:** The loss $J(\phi, \theta)$ for a randomly sampled training batch.

  $\mathcal{I} \leftarrow$ RANDOMSAMPLE$(\{1, \ldots, N\}, B)$              ▷ Select $B$ instance indices for batch
  $J_{TC}(\theta), J_{IC}(\phi) \leftarrow 0, 0$                    ▷ Initialize classification losses
  **for** $i$ in $\mathcal{I}$ **do**
    $\mathbf{z}_{v_i}, \mathbf{z}_{t_i} \leftarrow f_\phi(v_i), f_\theta(t_i)$                 ▷ Embed images and texts
    $p_I \leftarrow$ softmax$(\mathbf{W}_I \mathbf{z}_{v_i} + \mathbf{b}_I)$           ▷ Image classifier probabilities
    $p_T \leftarrow$ softmax$(\mathbf{W}_T \mathbf{z}_{t_i} + \mathbf{b}_T)$           ▷ Text classifier probabilities
    $J_{TC}(\theta) \leftarrow J_{TC}(\theta) + \frac{1}{B}$ crossentropy$(p_T, y_i)$   ▷ Text classification loss
    $J_{IC}(\phi) \leftarrow J_{IC}(\phi) + \frac{1}{B}$ crossentropy$(p_I, y_i)$    ▷ Image classification loss
  **end for**
  $J_{TR}(\phi, \theta), J_{IR}(\phi, \theta) \leftarrow 0, 0$                 ▷ Initialize retrieval losses
  **for** $i$ in $\mathcal{I}$ **do**

$$J_{TR}(\phi, \theta) \leftarrow J_{TR}(\phi, \theta) + \frac{1}{B}\left[ d(\mathbf{z}_{v_i}, \mathbf{z}_{t_i}) + \log \sum_{j \in \mathcal{I}} \exp(-d(\mathbf{z}_{v_i}, \mathbf{z}_{t_j})) \right] \quad \triangleright \text{Text retrieval loss}$$

$$J_{IR}(\phi, \theta) \leftarrow J_{IR}(\phi, \theta) + \frac{1}{B}\left[ d(\mathbf{z}_{v_i}, \mathbf{z}_{t_i}) + \log \sum_{j \in \mathcal{I}} \exp(-d(\mathbf{z}_{t_i}, \mathbf{z}_{v_j})) \right] \quad \triangleright \text{Image retrieval loss}$$

  **end for**
  $J(\phi, \theta) \leftarrow \lambda J_{TR}(\phi, \theta) + (1 - \lambda) J_{IR}(\phi, \theta)$      ▷ Add retrieval loss to the total loss
  $J(\phi, \theta) \leftarrow (1 - \kappa) J(\phi, \theta) + \frac{\kappa}{2}(J_{TC}(\theta) + J_{IC}(\phi))$   ▷ Add classification loss to the total loss

---

## 2.1 RETRIEVAL LOSS FUNCTION

Pairwise cross-modal loss function is based solely on the pairwise relationships between texts and images. We choose to use the metric learning approach to capture the relationship between images and texts. Now, suppose $d$ is a metric $d : \mathcal{Z} \times \mathcal{Z} \to \mathbb{R}^+$, $v_i$ is an image and $\tau = \{t_{j'}\}$ is a collection of arbitrary texts sampled uniformly at random, of which text $t_j$ belongs to $v_i$. We propose the following model for the probability of image $v_i$ and text $t_j$ to belong to the same object instance:

$$p_{\phi,\theta}(i = j | v_i, t_j, \tau) = \frac{\exp(-d(f_\phi(v_i), f_\theta(t_j)))}{\sum_{t_{j'} \in \tau} \exp(-d(f_\phi(v_i), f_\theta(t_{j'})))} . \tag{1}$$

The learning is then based on the following cross-entropy loss defined on the batch of size $B$:

$$J_{TR}(\phi, \theta) = -\frac{1}{B} \sum_{i,j=1}^{B} \ell_{i,j} \log p_{\phi,\theta}(i = j | v_i, t_j, \{t_{j'}\}_{j'=1}^{B}), \tag{2}$$

where $\ell_{i,j}$ is a binary indicator of the true match ($\ell_{i,j} = 1$, if $i = j$ and 0 otherwise). Note that the expression above has the interpretation of the text retrieval loss. It attains its smallest value when for each image in the batch we manage to assign probability 1 to its respective text and 0 to all other texts. This can be further expanded as:

$$J_{TR}(\phi, \theta) = \frac{1}{B} \sum_{i=1}^{B} \left( d(f_\phi(v_i), f_\theta(t_i)) + \log \left[ \sum_{t_{j'} \in \tau} \exp(-d(f_\phi(v_i), f_\theta(t_{j'}))) \right] \right). \tag{3}$$

Exchanging the order of image and text in the probability model (1) leads to the image retrieval loss, $J_{IR}(\phi, \theta)$. The two losses are mixed using parameter $\lambda \in [0, 1]$ as shown in Algorithm 1.

The pairwise retrieval loss functions are responsible for the modality alignment. In addition to those, we propose to include, as mentioned above, the usual image and text classifier losses. These losses are responsible for reducing the intraclass variability of representations. The classifier losses are added to the retrieval losses using a mixing parameter $\kappa \in [0, 1]$ as shown in Algorithm 1.

## 2.2 Balancing Accuracy on the Seen and Unseen Classes

Let us define class prototypes $\mathbf{p}(y)$, each based on the set of texts $\mathcal{T}_y$ belonging to class $y$, $\{\mathbf{p}(y) = \frac{1}{|\mathcal{T}_y|}\sum_{t_i \in \mathcal{T}_y} f_\theta(t_i) \mid y \in \mathcal{Y}\}$. In the context of GZSL, the nearest neighbor decision rule for a given image $v$ and its features $\mathbf{z}_v = f_\phi(v)$ has the following form:

$$\widehat{y} = \arg\min_{y \in \mathcal{Y}} d(\mathbf{z}_v, \mathbf{p}(y)) \ . \tag{4}$$

The most acute problem faced in this setup is the accuracy imbalance between seen and unseen classes. A very representative case clearly outlining the imbalance problem is presented in Table 6 of (Xian et al., 2018c), where accuracy on the seen classes is always significantly greater than the accuracy on unseen ones. In order to measure and control the imbalance, three metrics are commonly used to assess the classification performance in the GZSL scenario: the Top-1 accuracy on the seen categories ($\mathbf{s}$), the Top-1 accuracy on the unseen categories ($\mathbf{u}$) and their harmonic mean, $\mathbf{H} = \mathbf{u} \cdot \mathbf{s}/(\mathbf{u} + \mathbf{s})$. The main metric to assess GZSL performance is then $\mathbf{H}$, which quantifies both $\mathbf{u}$ and $\mathbf{s}$.

To formalize the problem, we first introduce $y_v$, the true class label of image $v$. Mathematically, the main GZSL pain point is that $\mathbb{P}\{\widehat{y} \in \mathcal{Y}^{tr}|y_v \in \mathcal{Y}^{ts}\}$ is significantly greater than $\mathbb{P}\{\widehat{y} \in \mathcal{Y}^{ts}|y_v \in \mathcal{Y}^{tr}\}$. In other words, the problem is that a given image is more likely to be confused with one of the seen classes if it belongs to an unseen class than vice versa. Our approach to solving the problem is based on the following probabilistic representation of the event space for the decision rule in Equation (4):

$$\mathbb{P}\{\widehat{y} \in \mathcal{Y}^{tr}|y_v \in \mathcal{Y}^{ts}\} = \mathbb{P}\left\{\min_{y \in \mathcal{Y}^{tr}} d(\mathbf{z}_v, \mathbf{p}(y)) < \min_{y \in \mathcal{Y}^{ts}} d(\mathbf{z}_v, \mathbf{p}(y)) \mid y_v \in \mathcal{Y}^{ts}\right\} \ . \tag{5}$$

Rephrasing, the most acute GZSL error happens when the prototype of one of the seen classes is closer to an image embedding from an unseen class than any of the prototypes of the unseen classes.

To rectify the situation we propose the following very direct solution to balance $\mathbb{P}\{\widehat{y} \in \mathcal{Y}^{tr}|y_v \in \mathcal{Y}^{ts}\}$ and $\mathbb{P}\{\widehat{y} \in \mathcal{Y}^{ts}|y_v \in \mathcal{Y}^{tr}\}$. We introduce a positive scalar $\alpha \in \mathbb{R}^+$ and scale all the distances corresponding to the seen prototypes by $1 + \alpha$. This gives rise to the following scaled distance $d_\alpha$:

$$d_\alpha(\mathbf{z}_v, \mathbf{p}(y)) = \begin{cases} (1+\alpha)d(\mathbf{z}_v, \mathbf{p}(y)), & \text{if } y \in \mathcal{Y}^{tr} \\ d(\mathbf{z}_v, \mathbf{p}(y)), & \text{otherwise} \end{cases} \ . \tag{6}$$

The misclassification between unseen as seen classes for the classifier $\widehat{y}_\alpha$, based on (6) is then:

$$\mathbb{P}\{\widehat{y}_\alpha \in \mathcal{Y}^{tr}|y_v \in \mathcal{Y}^{ts}\} = \mathbb{P}\left\{(1+\alpha)\min_{y \in \mathcal{Y}^{tr}} d(\mathbf{z}_v, \mathbf{p}(y)) < \min_{y \in \mathcal{Y}^{ts}} d(\mathbf{z}_v, \mathbf{p}(y))|y_v \in \mathcal{Y}^{ts}\right\} \ , \tag{7}$$

and it has the following property: for any $0 \leq \alpha_1 \leq \alpha_2$, $\mathbb{P}\{\widehat{y}_{\alpha_1} \in \mathcal{Y}^{tr}|y_v \in \mathcal{Y}^{ts}\} \geq \mathbb{P}\{\widehat{y}_{\alpha_2} \in \mathcal{Y}^{tr}|y_v \in \mathcal{Y}^{ts}\}$, *i.e.* $\mathbb{P}\{\widehat{y}_\alpha \in \mathcal{Y}^{tr}|y_v \in \mathcal{Y}^{ts}\}$ is a monotone non-increasing function of $\alpha$ and we can reduce it by increasing $\alpha$ (please refer to Appendix A for a proof). Consider now $\mathbb{P}\{\widehat{y}_\alpha \in \mathcal{Y}^{tr}|y_v \in \mathcal{Y}^{tr}\}$, which is a probability that we classify an image $v$ from one of the seen classes as still one of the seen classes. Using exactly the same chain of arguments as in Appendix A, it is straightforward to show that the probability is a non-increasing function of $\alpha$. Hence the probability $\mathbb{P}\{\widehat{y}_\alpha \in \mathcal{Y}^{ts}|y_v \in \mathcal{Y}^{tr}\} = 1 - \mathbb{P}\{\widehat{y}_\alpha \in \mathcal{Y}^{tr}|y_v \in \mathcal{Y}^{tr}\}$ is a non-decreasing function of $\alpha$. Therefore, as $\alpha$ increases, we expect more classification errors in classifying images from seen classes, because some of them will be classified as one of the unseen classes.

To sum up, given the arguments presented above we expect that by varying $\alpha > 0$ we can balance the error rate $\mathbb{P}\{\widehat{y}_\alpha \in \mathcal{Y}^{tr}|y_v \in \mathcal{Y}^{ts}\}$ of leaking the unseen class images into seen class classification decision and the error rate $\mathbb{P}\{\widehat{y}_\alpha \in \mathcal{Y}^{ts}|y_v \in \mathcal{Y}^{tr}\}$ of leaking the seen class images into unseen class classification decision. This is possible as we just showed above that $\mathbb{P}\{\widehat{y}_\alpha \in \mathcal{Y}^{tr}|y_v \in \mathcal{Y}^{ts}\}$ is a non-increasing function of $\alpha$, while $\mathbb{P}\{\widehat{y}_\alpha \in \mathcal{Y}^{ts}|y_v \in \mathcal{Y}^{tr}\}$ is a non-decreasing one. It is also important to emphasize that $\alpha$ is applied *only* to distances between the query embedding and the prototypes of seen classes and it is constant over seen classes. Therefore, the application of $\alpha$ does not at all affect the classification error rates either *within* $\mathcal{Y}^{tr}$ or *within* $\mathcal{Y}^{ts}$. Varying $\alpha$ balances exclusively the classification errors arising from transitions between seen and unseen class labels. We study the empirical aspects of balancing $\alpha$ in Section 4.4.

Table 1: Generalized zero-shot Top-1 classification accuracy.

|  | CUB | | | FLOWERS | | |
|---|---|---|---|---|---|---|
|  | **u** | **s** | **H** | **u** | **s** | **H** |
| CADA-VAE (Schönfeld et al., 2019) | n/a | n/a | 53.4 | n/a | n/a | n/a |
| f-CLSWGAN (Xian et al., 2018d) | 50.3 | 58.3 | 54.0 | 59.0 | 73.8 | 65.6 |
| f-VAEGAN-D2 (Xian et al., 2019) | 48.4 | 60.1 | 53.6 | 56.8 | 74.9 | 64.6 |
| cycle-(U)WGAN (Felix et al., 2018) | 47.9 | 59.3 | 53.0 | 61.6 | 69.2 | 65.2 |
| COSMO+f-CLSWGAN (Atzmon & Chechik, 2019) | n/a | n/a | n/a | 59.6 | 81.4 | 68.8 |
| CLAREL (Ours) | 59.3 | 52.6 | **55.8** | 73.0 | 73.6 | **73.3** |

Table 2: Zero-shot Top-1 classification accuracy.

|  | CUB | FLOWERS |
|---|---|---|
| CADA-VAE (Schönfeld et al., 2019) | n/a | n/a |
| f-CLSWGAN (Xian et al., 2018d) | 57.3 | 67.2 |
| f-VAEGAN-D2 (Xian et al., 2019) | 61.0 | 67.7 |
| cycle-(U)WGAN (Felix et al., 2018) | 58.6 | 70.3 |
| CLAREL (Ours) | **66.7** | **76.8** |

## 3 RELATED WORK

ZSL approaches aim at recognizing objects belonging to classes unseen during training (Larochelle et al., 2008; Palatucci et al., 2009). This has been extended to the GZSL framework in which the decision space consists of both seen and unseen classes (Socher et al., 2013; Xian et al., 2018c). The classical zero-shot approaches build a joint visual-semantic space, relying on a linear cross-modal compatibility function (e.g. dot-product between query embedding and semantic prototypes or a variation of a hinge loss) (Frome et al., 2013; Akata et al., 2015; 2016; Reed et al., 2016). Non-linear variants of the compatibility has also been explored (Xian et al., 2016; Socher et al., 2013). Extending previously proposed cross-modal transfer approaches based on auto-encoders (Hubert Tsai et al., 2017) and cross-domain learning (Gretton et al., 2007), more recent line of work (Schönfeld et al., 2019; Xian et al., 2018d; 2019; Felix et al., 2018; Verma et al., 2018) relies on combining these approaches and their variations with dataset augmentation tools such as GAN (Goodfellow et al., 2014) and VAE (Kingma & Welling, 2014). It is argued that the use of those tools helps to resolve one of the prominent problems in GZSL scenario: classifying images from unseen classes as one of the seen classes. There exist approaches that try to tackle this same problem via temperature calibration (Liu et al., 2018) originally proposed by Hinton et al. (2015). Chao et al. (2016); Das & Lee (2019) proposed an approach to seen/unseen accuracy balancing that is very similar to ours, based on heuristic arguments. We extend this line of work here by providing a probabilistic justification for the balancing effect observed when applying metric rescaling. Atzmon & Chechik (2019) propose a more sophisticated way to deal with seen/unseen imbalance via adaptive confidence smoothing and gating, yet as authors note it is much simpler to train than the existing GAN-based zero-shot approaches. In this work, we introduce arguably the simplest zero-shot representation training approach of all, and we demonstrate that when the image level text information is available, it achieves the state-of-the-art results on GZSL task on two well-known datasets.

## 4 EXPERIMENTAL RESULTS

### 4.1 DATASETS

We focus on learning embeddings for fine-grained visual descriptions and test them in ZSL/GZSL scenario. To test the quality of trained embeddings we focus on datasets that provide paired images and text descriptions, such as Caltech-UCSD-Birds (CUB) (Welinder et al., 2010) and Oxford Flowers (FLOWERS) (Nilsback & Zisserman, 2008), that were augmented with textual descriptions by Reed et al. (2016). We use the GZSL splits proposed by Xian et al. (2018c). The attribute-based datasets,

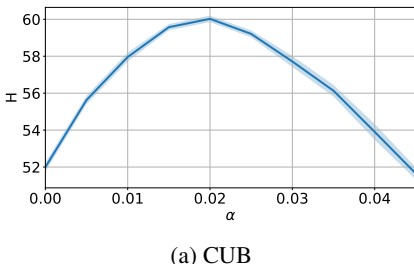
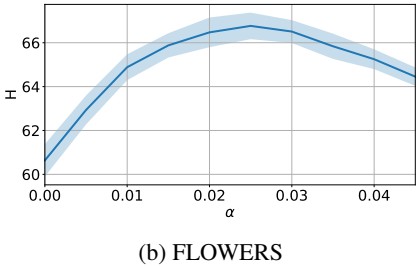

(a) CUB                               (b) FLOWERS

Figure 2: Harmonic mean Top-1 accuracy on seen and unseen, **H**, against the value of $\alpha$ on the validation set. The curves represent the mean and 95% confidence intervals over 10 optimization runs. Results are stable over different runs. **H** exhibits a distinct inverted U-shape w.r.t. $\alpha$.

such as SUN (Patterson et al., 2014) and AWA (Lampert et al., 2014) do not contain this information and do not have a notion of entity of a class in them. They are out of the scope of the current paper.

## 4.2 ARCHITECTURE AND TRAINING DETAILS

Below, we provide more detailed description of parameters used to build and train the architecture depicted in Figure 1. We use exactly the same hyperparameter settings for CUB and FLOWERS. The text feature extractor is built by cascading two ResNet blocks, followed by a BiLSTM. Each ResNet block has 3 convolutional/batch norm layers. The number of filters in the ResNet blocks is 128 and 256, BiLSTM has 512 filters for forward and backward branches (1024 total). All variables in the convolutional stack (including the batch normalization parameters $\gamma$ and $\beta$) are L2-penalized with weight 0.001. The image feature extractor is a ResNet-101 with fixed weights pretrained on the split of ImageNet proposed by Xian et al. (2018c). In this work we use precomputed image features, available in (Xian et al., 2018a) for CUB and in (Xian et al., 2018b) for FLOWERS. Image and text features are projected in the common embedding space of size 1024 with FC layers with no non-linearity. They are preceded with a dropout of 0.25. The trainable components of the model are trained for 150k batches of size 32 using SGD with initial learning rate of 0.1 that is annealed by a factor of 10 every 50k batches. For each batch, we sample 32 instances, each instance includes a vector of precomputed ResNet-101 features and 10 text descriptions corresponding to it, according to the original dataset definition Reed et al. (2016). All 10 text descriptions are processed via the CNN/LSTM stack and the resulting embeddings are average pooled to create a vector representation of length 1024.

## 4.3 KEY RESULTS

Our key empirical results are compared in Table 1 and in Table 2 against the latest state of the art. Our results are based on the settings of $\lambda = 0.5$, $\kappa = 0.5$ and $\alpha$ selected on the validation sets of CUB and FLOWERS datasets. Please refer to Section 4.5 for the analysis of stability with respect to the choices of $\lambda$ and $\kappa$ and Sections 2.2 and 4.4 for more details on the selection of $\alpha$. The combination of the proposed training method and the rebalancing of the metric space results in the state-of-the-art performance. Most of the current methods rely on the dataset augmentation techniques based on GANs, VAEs or combinations thereof. Those are clearly complementary w.r.t. our method and their addition to the training procedure is likely to further boost the performance of our proposed approach. However, this is outside of the scope of the current work. Moreover, the proposed method is state-of-the-art on FLOWERS even when compared against (Atzmon & Chechik, 2019) that uses both more sophisticated GAN based embedding learning approach and a more sophisticated seen/unseen error rate balancing based on COSMO. It is important to note that Atzmon & Chechik (2019) did not report the sentence level results on CUB. Yet, when applied on attributes together with f-CLSWGAN (Xian et al., 2018d) COSMO resulted in 0.8% performance drop and when applied with LAGO (Atzmon & Chechik, 2018) it achieved 0.5 % improvement over the attribute based state of the art.

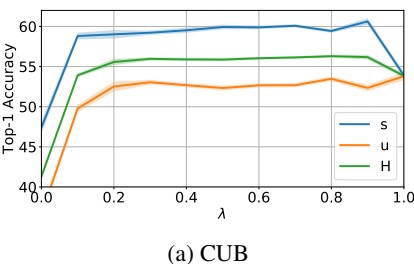
(a) CUB

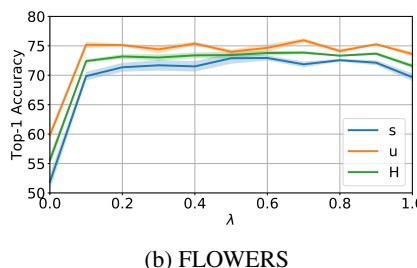
(b) FLOWERS

Figure 3: Harmonic mean Top-1 accuracy on seen and unseen, **H**, against $\lambda$, the relative weight of image and text retrieval loss terms. $\lambda = 0$ corresponds to the case of image retrieval loss having weight 1 and text retrieval loss having weight 0. Mean over 10 optimization runs.

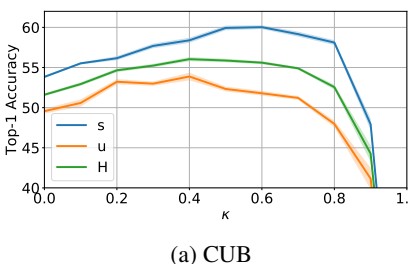
(a) CUB

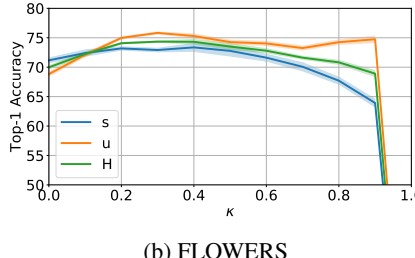
(b) FLOWERS

Figure 4: The plot of the harmonic mean Top-1 accuracy on seen and unseen, **H**, against $\kappa$, the relative weight of the retrieval and the classification loss terms. $\kappa = 0$ corresponds to the case of classification loss having weight 0. The curves represent the mean over 10 optimization runs.

### 4.4 ON THE SEEN/UNSEEN ACCURACY BALANCING

Figure 2 demonstrates the plot of harmonic mean Top-1 accuracy, **H**, against the value of $\alpha$ on the validation sets of CUB and FLOWERS datasets. The validation set is constructed by further splitting the train set on both datasets. For example, CUB has a train set of 5875 images from 100 seen classes and a validation set of 2946 images from 50 unseen classes. We further divide the train set into 4700 train images from 100 seen classes, 1175 seen validation images (4700 + 1175 = 5875) and we use all the 2946 images from 50 classes as the unseen validation set. Once the value of $\alpha$ is determined we train the representation on the full train+val subset and report results on the test split (the usual practice in GZSL). We confirm on the validation set that **H** exhibits an inverted U-shape behavior as a function of $\alpha$, which was theoretically predicted in Section 2.2. Therefore, $\alpha$ can be selected on the validation set and then applied to re-scale the metric space to balance the accuracy on seen and unseen classes during test time as described in Section 2.2.

### 4.5 ABLATION STUDY

Figure 3 presents the results of the ablation study on the importance of image and text retrieval losses. We see that all of the Top-1 accuracies (**H**, **s**, **u**) are stable in the range of $\lambda \in [0.2, 0.9]$, when both losses have tangible weight. Removing either text or image retrieval losses (setting $\lambda$ to 0 or 1 respectively) leads to performance drop in both cases. Removing the text retrieval loss (case $\lambda = 0$) results in the most significant drop. This is due to the fact that the text retrieval task is more tightly related to the GZSL task. At the batch level, retrieving the right text given an image is equivalent to identifying the correct class encoded by a text prototype during ZSL inference step. The image retrieval task is not directly related to solving the ZSL problem and yet it does yield a positive regularizing effect on both CUB and FLOWERS.

Figure 4 shows the results of the ablation study of the interplay between the retrieval loss and the classification loss. We observe, just as in the case with $\lambda$, that there exists a reasonably flat and stable range of $\kappa \in [0.2, 0.6]$. The range for $\kappa$ is a bit smaller. $\kappa = 1$ results in the catastrophic performance

Table 3: Generalized zero-shot Top-1 classification accuracy, ablation study.

| | | | CUB | | | FLOWERS | | |
|---|---|---|---|---|---|---|---|---|
| $\alpha$ | $\lambda$ | $\kappa$ | **u** | **s** | **H** | **u** | **s** | **H** |
| 0.0 | 0.5 | 0.5 | 38.3 | 65.3 | 48.3 | 55.1 | 84.6 | 66.7 |
| 0.0 | 0.5 | 0.0 | 39.3 | 57.5 | 46.7 | 54.0 | 78.1 | 63.8 |
| ✓ | 0.5 | 0.0 | 53.8 | 49.6 | 51.6 | 71.7 | 67.2 | 69.4 |
| ✓ | 0.0 | 0.5 | 47.4 | 36.6 | 41.3 | 51.5 | 60.5 | 55.6 |
| ✓ | 1.0 | 0.5 | 53.9 | 53.8 | 53.8 | 69.5 | 73.9 | 71.6 |
| ✓ | 0.5 | 0.5 | 59.3 | 52.6 | **55.8** | 73.0 | 73.6 | **73.3** |

drop: the classification losses by themselves do not enforce any modality alignment (please refer to Fig. 1 and Algorithm 1 clearly demonstrating this).

Table 3 studies the effects of different loss terms on the harmonic mean Top-1 accuracy **H**. The best result is achieved when all loss terms are active and when the metric space rescaling is on (the case of $\lambda = 0.5$, $\kappa = 0.5$ and $\alpha$ is checked, the last line in the table). Comparing this with the case when there is no metric space rescaling (first line with $\alpha = 0$), we see that the rescaling helps to decrease the gap between seen and unseen classification accuracy. For CUB, the discrepancy reduction is from around 30% to around 6%, for FLOWERS it is from around 30% to around 1%. We would like to stress that we only use images and texts from the training set to achieve that. Going to the second line in the table (the image/text classification loss is inactive, $\kappa = 0$) and comparing it to the first one, we assess the effect of the image/text classification loss. It barely affects the performance on unseen set, but it significantly boosts the classification accuracy on the seen set (around 8% on both datasets). This is logical: adding a classifier loss results in a better classifier of the test images from the seen classes. This alone does not make it a better GZSL classifier, however. Only when applied together with metric space rescaling, this results in the performance boost (please refer to lines 1 and 6 in Table 3). Our interpretation is that the addition of the image/text classifier loss helps to reduce the intraclass variability in embeddings and provides for tighter clustering. However, this also leads to overfit on the classification task. This is accounted for by metric rescaling that enables the learnings from the image/text classification task be transferred effectively into the GZSL task.

The comparison of the last four rows of Table 3 leads us to believe that all the proposed loss terms outlined in Fig. 1 and Algorithm 1 are important for achieving the state-of-the-art performance. Excluding any one of them (corresponding to the extreme values $\lambda = 0$, $\lambda = 1$, $\kappa = 0$) leads to performance deterioration. Finally, an interesting observation can be made by comparing line 3 of Table 3 with performance of algorithms in Table 1. In this case our algorithm does not use any class labels and relies on training using exclusively the retrieval losses that can be calculated only based on the pairwise relationships between texts and images. We can see that using this type of supervision alone already results in a very high-quality representation. The representation is competitive against the latest GAN/VAE based approaches on CUB and is state-of-the-art on FLOWERS. This opens up new exploration avenues showing that in the case when very fine-grained modality outputs are available (image and text description pairs being a very prominent example), the high-quality representations may be learned without relying on manually supplied class labels.

## 5 CONCLUSIONS

We propose and empirically validate two improvements to the process of learning fine-grained cross-modal representations. First, we confirm the hypothesis that in the context of paired images and texts, a deep metric learning approach can be driven by an instance-based retrieval loss resulting in competitive generalized zero shot classification results. Combined with an additional class label based image/text crossentropy term this results in state-of-the-art performance on two well known datasets, CUB and FLOWERS. This is an interesting result demonstrating that high-quality deep representations can be trained relying largely on pairwise relationships between modalities. On top of that, we propose a solution to one of the prominent problems in GZSL: classifying instances of unseen classes as seen ones. We mathematically analyze and empirically validate the method of adjusting a single scalar that transcends in its effectiveness advanced dataset augmentation and training approaches based on GANs and VAEs.

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

## A  THE ANALYSIS OF ERROR RATES

We show that $\mathbb{P}\{\widehat{y} \in \mathcal{Y}^{tr}|y_v \in \mathcal{Y}^{ts}\} \geq \mathbb{P}\{\widehat{y}_\alpha \in \mathcal{Y}^{tr}|y_v \in \mathcal{Y}^{ts}\}$. Let us define $\delta_{tr} \equiv \min_{y \in \mathcal{Y}^{tr}} d(\mathbf{z}_v, \mathbf{p}(y))$ and $\delta_{ts} \equiv \min_{y \in \mathcal{Y}^{ts}} d(\mathbf{z}_v, \mathbf{p}(y))$, then Equation (7) can be rewritten as:

$$\mathbb{P}\{\widehat{y}_\alpha \in \mathcal{Y}^{tr}|y_v \in \mathcal{Y}^{ts}\} = \mathbb{P}\left\{(1+\alpha)\delta_{tr} < \delta_{ts}|y_v \in \mathcal{Y}^{ts}\right\} . \tag{8}$$

Let us consider the probability of event $\delta_{tr} < \delta_{ts}$ and decompose it as follows:

$$\mathbb{P}\left\{\delta_{tr} < \delta_{ts}|y_v \in \mathcal{Y}^{ts}\right\} = \mathbb{P}\left\{(1+\alpha)\delta_{tr} < (1+\alpha)\delta_{ts}|y_v \in \mathcal{Y}^{ts}\right\}$$
$$= \mathbb{P}\left\{(1+\alpha)\delta_{tr} < \delta_{ts} \cup \delta_{ts} \leq (1+\alpha)\delta_{tr} < (1+\alpha)\delta_{ts}|y_v \in \mathcal{Y}^{ts}\right\}$$
$$= \mathbb{P}\left\{(1+\alpha)\delta_{tr} < \delta_{ts}|y_v \in \mathcal{Y}^{ts}\right\} + \mathbb{P}\left\{\delta_{ts} \leq (1+\alpha)\delta_{tr} < (1+\alpha)\delta_{ts}|y_v \in \mathcal{Y}^{ts}\right\}$$
$$- \mathbb{P}\left\{(1+\alpha)\delta_{tr} < \delta_{ts} \cap \delta_{ts} \leq (1+\alpha)\delta_{tr} < (1+\alpha)\delta_{ts}|y_v \in \mathcal{Y}^{ts}\right\}$$
$$= \mathbb{P}\left\{(1+\alpha)\delta_{tr} < \delta_{ts}|y_v \in \mathcal{Y}^{ts}\right\} + \mathbb{P}\left\{\delta_{ts} \leq (1+\alpha)\delta_{tr} < (1+\alpha)\delta_{ts}|y_v \in \mathcal{Y}^{ts}\right\} .$$

The transitions are based on the relationship between probabilities of arbitrary events $A$ and $B$, $\mathbb{P}\{A \cup B\} = \mathbb{P}\{A\} + \mathbb{P}\{B\} - \mathbb{P}\{A \cap B\}$, and in our case $\mathbb{P}\{A \cap B\} = 0$. This implies that:

$$\mathbb{P}\{\widehat{y}_\alpha \in \mathcal{Y}^{tr}|y_v \in \mathcal{Y}^{ts}\} = \mathbb{P}\{\widehat{y} \in \mathcal{Y}^{tr}|y_v \in \mathcal{Y}^{ts}\} - \mathbb{P}\left\{\frac{\delta_{ts}}{(1+\alpha)} \leq \delta_{tr} < \delta_{ts}|y_v \in \mathcal{Y}^{ts}\right\}$$

$$\leq \mathbb{P}\{\widehat{y} \in \mathcal{Y}^{tr}|y_v \in \mathcal{Y}^{ts}\}. \tag{9}$$

We have just shown that for a non-negative $\alpha$ the probability of misclassifying an image from an unseen class as one of the seen classes is smaller for the decision rule $\widehat{y}_\alpha$ than for the original decision rule $\widehat{y}$. In fact, we can make a stronger claim. Since $\delta_{ts}$ and $\delta_{tr}$ are non-negative, it is clear that the length of interval $[\delta_{ts}/(1+\alpha), \delta_{ts})$ increases as $\alpha$ increases, and hence probability that $\delta_{tr}$ falls in this interval is non-decreasing with increasing $\alpha$. Thus we have for any $0 \leq \alpha_1 \leq \alpha_2$, $\mathbb{P}\{\widehat{y}_{\alpha_1} \in \mathcal{Y}^{tr}|y_v \in \mathcal{Y}^{ts}\} \geq \mathbb{P}\{\widehat{y}_{\alpha_2} \in \mathcal{Y}^{tr}|y_v \in \mathcal{Y}^{ts}\}$, *i.e.* $\mathbb{P}\{\widehat{y}_\alpha \in \mathcal{Y}^{tr}|y_v \in \mathcal{Y}^{ts}\}$ is a monotone non-increasing function of $\alpha$ and we can reduce it by increasing $\alpha$.

