# OpenReview forum: "CLAREL: classification via retrieval loss for zero-shot learning"
_ICLR.cc/2020/Conference — Reject_

### Official Review · AnonReviewer2 · 2019-10-23
**Official Blind Review #2**

**Rating:** 6

**Review:**

This paper presents two main contributions: first, a simple retrieval-based objective for learning joint text and image representations, and second, a metric-scaling term that improves performance on unseen classes in the generalized zero-shot (GZSL) setting. They evaluate on two datasets, CUB and FLOWERS, consisting of images paired with text descriptions, and show that their relatively simple technique outperforms complex GAN and VAE-based models.

Weak accept, see considerations below. The results seem solid, and the architecture is quite simple compared to other models on this task. However, there are some gaps in the analysis, and the paper would benefit from a more careful comparison to the literature.

The results are strong, and it’s encouraging to see a simple approach like this outperform more complex models. However, it’s not clear how novel the retrieval model is; as implemented, it’s similar to a sampled softmax or noise contrastive estimation - the paper would benefit from a more careful comparison of their approach to established alternatives, and a discussion of why this approach should learn better representations than a VAE or GAN. It’s also not clear how novel the metric scaling technique is. It undeniably works well, but the use seems remarkably similar to (Das & Lee 2019, Section II.D).

The observation (Section 4.5) that the retrieval-only model performs well without any class labels is interesting. How does this compare to other unsupervised approaches? It also appears that the version with the metric scaling here still performs better - is this because the retrieval model is trained only on examples from the “seen” classes? Does this gap go away if (unlabeled) examples from the unseen classes are available at training time?

A few other notes:
* Section 4.5: “Our interpretation is that the addition of the image/text classifier loss helps to reduce the intraclass variability in embeddings and provides for tighter clustering.” - could you test this directly, or provide some visualization, such as t-SNE plots?
* The image representations are initialized from an ImageNet model, but text is trained from scratch. Why not use a pretrained text encoder like BERT?
* Table 3: it would help to have text labels for these rows, or a reminder in the caption of what \lambda = 0 and \kappa = 0 mean (otherwise, one needs to flip/scroll back to page 4)
* Figure 2: what are the actual values of u and s that contribute to this plot?


**Experience Assessment:**

I have read many papers in this area.

**Review Assessment: Checking Correctness Of Derivations And Theory:**

I assessed the sensibility of the derivations and theory.

**Review Assessment: Checking Correctness Of Experiments:**

I assessed the sensibility of the experiments.

**Review Assessment: Thoroughness In Paper Reading:**

I read the paper at least twice and used my best judgement in assessing the paper.

---

> ### Author Response · Authors · 2019-11-15
> **Response to Reviewer 2**
>
> We would like to sincerely thank Reviewer 2 for providing very insightful and constructive feedback. All the points raised by the reviewer are well taken and will be used to strengthen the revised manuscript by clarifying the novelty of the work and by extending the related work section by adding the literature on contrastive training and sampled softmax. Detailed response follows below.
>
> The literature review will be extended as outlined in our response to Reviewer 3, plus the discussion to the sampled softmax. We find the parallel that Reviewer 2 drew between the sampled softmax and our work to be pretty interesting. The approach of Das & Lee 2019 is equivalent to metric scaling and our contribution here is a more detailed probabilistic analysis of the method. The results in Table 3 imply that good representations can be learned via the proposed composite loss and even only based on the retrieval loss. The existing GAN and VAE approaches use pairwise information available in the datasets only to construct a prototype for each class. In our view, this unnecessarily constraints the representation space by limiting the number of anchor points. In our case, the resulting embedding space is less constrained and it does not have to overfit its structure to the relatively small number of training set class prototypes to minimize the training loss. We are very interested to look at the actual structure of the space via t-SNE, as suggested by Reviewer 2. We will definitely include this analysis for different combinations of class- and image-level supervision in the next revised version (unfortunately, due to time constraints we will not be able to run this analysis before the end of the rebuttal period).
>
> Yes, the retrieval-only model with scaling is better than the one without scaling. It is true that the gap is due to the model not being exposed to unseen classes at train time. We have not studied the effect of adding the unseen classes in the training set. However, it is quite reasonable to assume that the more examples of the previously unseen classes are added to the train set, the smaller the gap will become.
>
> Response to a few other notes.
>
> 1. Good point, the t-SNE plots will be added, as discussed above.
> 2. We made sure that our train and evaluation setup is as close as possible to the original one defined by Reed et al. 2016, since it is a default. This is to ensure the apple-to-apple comparison. BERT will likely improve the scores, but then we would have to reimplement all the approaches in the literature as we would not be able to cite the currently published results.
> 3. This is a good point we will add a reminder in the caption to make sure the table is easy to read
> 4. We will add the u and s to the plot in Figure 2, as requested by the Reviewer

---

### Official Review · AnonReviewer3 · 2019-10-23
**Official Blind Review #3**

**Rating:** 3

**Review:**

The paper proposes to to use four different losses to train a joint text-image embedding space for zero-shot learning. The four losses consist of a classification loss given text descriptions, a classification loss given images, two contrastive losses given pairs of text and images. The paper also discusses how to balance seen and unseen classes, and it seems that, empirically, embeddings for seen classes are closer together while the embeddings for the unseen classes are further apart. A scaling factor that makes sure these distances are comparable for seen and unseen classes is introduced, and the paper gives an explanation for such a scaling. The final performance on the CUB and FLOWERS data set is impressive.

I am giving a score of 3. The engineering effort in the paper is appreciated, but the novelty is lacking. The losses introduced in the paper, broadly classified as contrastive loss, has been around since (Frome et al., 2013). See also (Karpathy et al., 2014; Wang et al., 2016) and in particular (Xian et al., 2016). The subtle differences between all these losses are whether there is a margin, whether the loss is smooth (i.e., a softmax as opposed a max), and how the negative samples are selected. I am surprised that the loss function being used by many are considered a contribution in the paper. The property of metric scaling is certainly interesting, but it does not answer why metric scaling is needed in the first place. Overall, the novelty is limited.

Below are some minor points and questions for the paper.

... anchor embeddings learned in one modality as prototypes ...
--> the word "prototype" is used extensively in the rest of the paper. i know this is a common term in the zero-shot learning community, but it might be good to give a formal definition early or at least give an informal definition.

Algorithm 1
--> what is the choice of d in the experiments?

... the probability of image v_i and text t_j to belong to the same object distance
--> isn't this the constrastive loss? what does it mean for the two to belong to the same object distance?

equations (2)
--> it seems that v_i and t_i are paired, but it is unclear from the text.

equation (1) and (2)
--> have you considered other (distributions of) negative samples?

... P{\hat{y} \in Y^{tr} | y_v \in Y^{ts}} is significantly greater than P{\hat{y} \in Y^{tr} | y_v \in Y^{tr}}
--> what's the definition of P{\hat{y} | y_v}? the notation is confusing and hinders the understanding of the rest of the discussion. it's probably easier to just talk about distances and use the nearest neighbor argument.

Figure 2
--> what's different between the 10 different runs? random seeds?

Figure 3
--> why is the performance significantly lower when lambda=0? can this be related to how negative samples are selected?

References:

DeViSE: a deep visual-semantic embedding model
A. Frome, G. S. Corrado, J. Shlens, S. Bengio, J. Dean, M. Ranzato, T. Mikolov
Neurips, 2013

Deep fragment embeddings for bidirectional image sentence mapping
A. Karpathy, A. Joulin, F. Li
Neurips, 2014

Learning deep structure-preserving image-text embeddings
L. Wang, Y. Li, S. Lazebnik
CVPR, 2016

Latent embeddings for zero-shot classification
Y. Xian, Z. Akata, G. Sharma, Q. Nguyen, M. Hein, B. Schiele
CVPR, 2016


**Experience Assessment:**

I have read many papers in this area.

**Review Assessment: Checking Correctness Of Derivations And Theory:**

I carefully checked the derivations and theory.

**Review Assessment: Checking Correctness Of Experiments:**

I carefully checked the experiments.

**Review Assessment: Thoroughness In Paper Reading:**

I read the paper thoroughly.

---

> ### Author Response · Authors · 2019-11-15
> **Response to Reviewer 3**
>
> We would like to sincerely thank Reviewer 3 for providing very insightful and constructive feedback. All the points raised by the reviewer are well taken and will be used to strengthen the revised manuscript by clarifying the novelty of the work and by providing additional details about definitions and interpretations of results. Detailed response follows below.
>
> The references provided by Reviewer 3 are related to our work, however they are not focused on addressing the same problem. First, Frome et al., 2013 and Xian et al., 2016 focus on learning zero-shot compatible representations based on the class-level supervision. They provided very important contributions to advance the state-of-the-art in zero-shot learning at the time they were published. However, they did not target to answer the same question as our current work. The question that we target to answer in our work is whether the image-level supervision in the text/image paired datasets (CUB and FLOWERS being good examples of such) can be used effectively to learn meaningful zero-shot capable representations? And if so, what is the relationship between the value provided by supervision signals at class-level and image level? Second, Karpathy et al., 2014 and Wang et al., 2016 focus on the problem of learning representations in the image/text pairing contexts, but these works clearly fall outside of the zero-shot domain. No experiments in these works have quantitatively demonstrated the ability to generalize beyond seen classes using their respective approaches. For example, Karpathy et al., 2014 heavily rely on image fragment detections by a RCNN pretrained on the full ImageNet and then fine-tuned on ImageNet Detection classes. In Section 4.3,  they state: "The detection CNN is trained to predict one of 200 ImageNet Detection classes, so it is not clear if the representation is powerful enough to support learning of more complex attributes of the objects or generalize to novel classes." Then they present only qualitative results (a few pictures) and conclude: "Qualitative results shown in Figure 5 suggest that the model is indeed capable of generalizing to more fine-grained subcategories (such as “black dog”, “soccer ball”) and to out of sample classes such as “rocky terrain” and “jacket”." However, no further conclusions about zero-shot capability of their method supported by qualitative results are provided in the paper (Karpathy et al., 2014). Wang et al., 2016 follow a very similar setup, but also additionally rely on bounding box labels at train time: "For our first experiment, we train our embedding without negative mining, using the same positive region-phrase pairs as CCA". As a conclusion, Karpathy et al., 2014 and Wang et al., 2016 are focusing on the region-based sentence/image alignment, relying on region based bounding box labelling supervision signals and/or pretrained RCNN, they did not focus on studying zero-shot transferability of learned embeddings. Also, Karpathy et al., 2014 and Wang et al., 2016 did not provide any empirical evidence to compare the performance and interplay between the image-level and the class-level supervision, unlike our work.
>
> We believe that our results are complementary and novel with respect to the previous work pointed out by Reviewer 3. We propose a new composite loss that enables more flexible handling of both image-level and class-level supervision signals. We empirically demonstrate on two datasets that the image-level supervision signals are effective and that they can be fruitfully combined with the class-level supervision signals to learn high-quality representations in the joint text/image space domain. We demonstrated that the learned representations are transferrable both in the zero-shot and in the generalized zero-shot evaluation scenarios.
>
> Answers to minor points.
>
> 1. good point, we will formally define a prototype in the revised draft
> 2. The choice of d in experiments is Euclidean, we will clarify this
> 3. $v_i$ and text $t_j$ belong to the same instance, if text $t_j$ describes image $v_i$, we will clarify this in the text
> 4. Indeed, $v_i$ and $t_i$ are paired, we will clarify it in the text
> 5. We have not considered other distributions of negative samples. It is likely that shaping the distribution may help, along with hard negative mining. However, we targeted to simplify the training procedure maximally to make its implementation as easy as possible in practical settings
> 6. We will look into ways of simplifying the definition of $P\{\hat{y} | y_v\}$ as suggested by the reviewer
> 7. Yes, the difference between the runs is random seed
> 8. At the batch level, retrieving the right text given an image is equivalent to identifying the correct class encoded by a text prototype during ZSL inference. Thus the text retrieval task is more tightly related to the GZSL task, as opposed to the image retrieval task.  Therefore, when the latter is the only loss left ($\lambda=0$), performance drops.

---

> > ### Comment · AnonReviewer3 · 2019-11-15
> > **score unchanged**
> >
> > > The question that we target to answer in our work is whether the image-level supervision in the text/image paired datasets (CUB and FLOWERS being good examples of such) can be used effectively to learn meaningful zero-shot capable representations?
> >
> > So it is safe to interpret that the contribution of the paper is that embeddings learned from a contrastive loss (be it hard or soft) can be used for zero-shot learning. I agree this is clearly demonstrated in the paper.
> >
> > > And if so, what is the relationship between the value provided by supervision signals at class-level and image level?
> >
> > I do not think the paper provides any satisfactory answer to this question. I do not see the value provided by supervision signals defined anywhere in the paper, unless it is just a different way of saying the performance metric. I also do not see a discussion on the relationship.
> >
> > > Second, Karpathy et al., 2014 and Wang et al., 2016 focus on the problem of learning representations in the image/text pairing contexts, but these works clearly fall outside of the zero-shot domain. No experiments in these works have quantitatively demonstrated the ability to generalize beyond seen classes using their respective approaches.
> >
> > This basically echos the first point: people have obtained embeddings trained with contrastive loss, but this has not been investigated for zero-shot learning.
> >
> > Judging from the reply, the only implication of the paper is that contrastive loss is suitable for learning embeddings for zero-shot tasks.
> >
> > I must say that this contribution is shallow. However, I would give a higher score if the paper could provide experiments, analyses, and some hypotheses to say why this is the case. I have read the reviews and comments, and it seems that other reviewers have similar concerns.
> >
> > To conclude, I stand by the score.

---

### Official Review · AnonReviewer1 · 2019-10-23
**Official Blind Review #1**

**Rating:** 1

**Review:**

This paper tackles zero-shot and generalised zero-shot learning by using the per-image semantic information. An instance-based loss is introduced to align images and their corresponding text in the same embedding space. To solve the extreme imbalanced issue of generalized zero-shot learning, the authors propose to scale the prediction scores of seen classes by a constant factor. They demonstrate technic contributions on CUB and Flowers datasets and the results achieve the state-of-the-art.
This paper should be rejected because (1) motivation is not well justified. This paper fails to convince me that it is practical to use per-image semantic information. Zero-shot learning aims to reduce the cost of annotation, but the per-image semantic information used in this paper conversely increases the annotation efforts. More specifically, this paper directly applies the per-image text descriptions proposed by Reed et al. '16, in which they annotated each image of CUB and Flowers by 10 different sentences. Due to the lack of such expensive annotations, this paper can be only evaluated on CUB and Flowers datasets,  results on other popular zero-shot learning datasets e.g., AWA, SUN, ImageNet, are essentially missing.
(2) novelty is limited. Using per-image semantic information is not new in zero-shot learning at all. Reed et al. '16 studied this problem and showed that per-image text-description can surpass per-image attributes in zero-shot learning. The loss function of this paper is also similar to  Reed et al. '16, which used the max-margin loss to align image and text pairs, v.s. the cross-entropy loss of this paper. Claiming that they are the first to use per-image semantic information in GZSL is not convincing because GZSL was not introduced at the time of Reed et al. '16. Metric scaling is not novel either. It is, in fact, equivalent to the calibration technic proposed by Chao et al. '16. The theory of metric scaling is redundant in my point of view because it is obvious that rescaling decreases the seen class scores.
(3) experiments are insufficient to support the contribution. Their experiments are not comparing apple to apple. Specifically, the competing methods are using per-class semantic information while their approach uses per-image semantic information which is unfair because per-image semantic information includes much more supervision. The authors should have compared the results of those methods trained with per-image semantic information. I would expect all the approach will benefit from per-image side information.



Post-rebuttal comments:  In the author responses, the authors have written long stories to fight against my reviews. But unfortunately,  a large part of the responses is not addressing my concerns. The authors repeatedly argue that their main contribution is to show that image-level supervision is an effective way to tackle (generalized) zero-shot learning. However, this contribution is not significant because Reed et al. '16 has demonstrated this point in the scenario of zero-shot learning. Extending this idea to generalized zero-shot learning is not a sufficient contribution for ICLR.


**Experience Assessment:**

I have published in this field for several years.

**Review Assessment: Checking Correctness Of Derivations And Theory:**

I carefully checked the derivations and theory.

**Review Assessment: Checking Correctness Of Experiments:**

I carefully checked the experiments.

**Review Assessment: Thoroughness In Paper Reading:**

I read the paper thoroughly.

---

> ### Author Response · Authors · 2019-11-15
> **Response to Reviewer 1, point (3)**
>
> In our experimental setup we are given access to exactly the same information as the other methods. Therefore, we are confident that the setup is fair and we explain this in detail below.
>
> Images and texts tend to naturally appear in pairs. This applies to many interesting realistic scenarios including the web based datasets, as well as the smaller scale datasets like FLOWERS and CUB. This is natural, because modality pairing is a lower level and easier to obtain information that is often available "for free". The higher level information then appears at the level of more abstract notions, for example 'class'. In the case of FLOWERS and CUB, each initial image was augmented with both text descriptions and a class label. The association of texts with a given class label is achieved $\textrm{only}$ via known pairwise relationship between a text and its respective image. All the methods that have been applied on the sentence based versions of CUB and FLOWERS used the summation of embeddings of all texts belonging to a given class as its prototype. However, such class prototypes in this context are impossible to construct without access to known pairwise image/text associations. Therefore, all of the methods applied on the sentence based versions of CUB and FLOWERS already did use pairwise image/text pairing information. Since this fact is not exposed on the surface, it is relatively easy to overlook, but it is important to take into account. Once this is taken into account, the fact that we use exactly the same information as the other methods appearing in Tables 1 and 2 becomes clear. Therefore, our evaluation framework is fair.
>
> Now, there is still another question. Did all the methods applied to the sentence based CUB and FLOWERS use the information that they are exposed to effectively? In other words, is the best way of using the available per-image semantic information to simply construct a class prototype using image/text pairs and class labels, like the other methods did? Definitely not, and this is exactly what our paper is about! This exactly what we claim as a contribution: "We demonstrate that the class-level information is important, but in the fine-grained text/image pairing scenarios, most of the GZSL accuracy can be extracted from the instance-based retrieval loss. To the best of our knowledge, this type of training has not been used in the GZSL literature."
>
> We are confident that we managed to convincingly demonstrate that there are better ways to use the available pair-wise text/image information for learning a high quality representation in joint text/image embedding space.  We proposed a concrete and very simple recipe to achieve this and we evaluated it on two different datasets consistently demonstrating that it works well.
>
> Now, Reviewer 1 also asks us to extend the methods existing in the literature with what we propose in this paper. We believe that what we included in the current manuscript already convincingly proves the point that the retrieval loss based training is effective for training meaningful zero-shot representations in the context of our study. This is exactly why we had pointed out in our original submission that what Reviewer 1 asked is outside of the scope of our current work:  "Most of the current methods rely on the dataset augmentation techniques based on GANs, VAEs or combinations thereof. Those are clearly complementary w.r.t. our method and their addition to the training procedure is likely to further boost the performance of our proposed approach. However, this is outside of the scope of the current work." However, this is definitely a very interesting direction for future work, along with studying zero-shot learning in the context of larger datasets. We hope that our work will draw attention to this promising research direction and will help to generate new exciting results in this research domain.
>
> We believe that we demonstrated that our experimental framework is sound and fair. Therefore, Reviewer's 1 point (3) is not a valid reason for reject.

---

> ### Author Response · Authors · 2019-11-15
> **Response to Reviewer 1, point (2)**
>
> We are convinced that Reviewer's 1 claim "The loss function of this paper is also similar to  Reed et al. '16, which used the max-margin loss to align image and text pairs" is false. We would like to point out that Reed et al. 2016 used class level supervision at train time as is obvious from equations (1), (3)-(7) https://arxiv.org/pdf/1605.05395.pdf, where class label is always present. The loss terms proposed by Reed et al. 2016 did not align image/text pairs, at all. Instead, they aligned an instance of text with a prototype of images averaged over a sample from a given class and vice versa, as evident from equations (3) and (4). Without class level labels, the losses proposed by Reed et al. 2016 are undefined and therefore cannot be used to train on image level datasets that contain only image/text pairings. One of the goals of our paper is actually to show that we can go beyond that.  Our proposed approach is very flexible in that it can use both image/text pairing information as well as class information, if such information is available. We clearly demonstrated in Table 3, line 3; that even without access to class labels during embedding training time, the quality of resulting embeddings learned simply from pairwise relationships between images and texts is very high. We are convinced that this is an interesting result that provides a motivation to continue work in this direction by embracing web scale datasets and perhaps even targeting to prove that the zero-shot way of learning representations may be a more promising venue than the reliance on manually labelled datasets like Imagenet.
>
> On top of that, the reviewer's claim about us allegedly "Claiming that they are the first to use per-image semantic information in GZSL" is false. We would like to point out that we only claimed in our contributions that "We demonstrate that the class-level information is important, but in the fine-grained text/image pairing scenarios, most of the GZSL accuracy can be extracted from the instance-based retrieval loss. To the best of our knowledge, this type of training has not been used in the GZSL literature." Indeed, we are not aware of results in the literature that would demonstrate an equivalent of our Table 3, where it is evident that instance-based retrieval loss is almost as effective as class-based loss.
>
> Next, Reviewer 1 claims "GZSL was not introduced at the time of Reed et al. '16". We argue that this claim is false too. The GZSL setup was considered already by Socher et al. 2013 and by the way we clearly gave credit to this publication in our literature review (it is one of our references). So GZSL setup was introduced at least 3 years before Reed et al. '16. In particular, Socher et al. 2013 defined the problem of constructing a classifier on joint space of seen and unseen classes (exactly the GZSL framework) as follows: "In general, we want to predict $p(y|x)$, the conditional probability for both seen and unseen classes $y \in Y_s \cup  Y_u$ given an image from the test set $x \in X_t$." Please refer to https://arxiv.org/pdf/1301.3666.pdf , section 5. On top of that, Socher et al. 2013 even proposed a solution to the seen/unseen accuracy imbalance problem that is very similar in nature to some of the latest work published in CVPR this year (Atzmon and Chechik 2019)
>
> Another claim by Reviewer1: "Metric scaling is not novel either. It is, in fact, equivalent to the calibration technic proposed by Chao et al. '16." If Reviewer 1 implies that we claimed in our text that metric scaling is novel then this claim is false. In our contributions we clearly stated: "Previous work proposed to use a heuristic trick, calibrated stacking (Chao et al., 2016) or calibration (Das & Lee, 2019), to solve the problem. We refer to this technique as metric rescaling in our work, and provide a sound probabilistic justification for it."
>
> Finally, "The theory of metric scaling is redundant in my point of view because it is obvious that rescaling decreases the seen class scores. ". First of all, if metric rescaling only decreased the seen class scores it would not solve the imbalance problem. The rescaling should also increase, or at least not decrease the unseen class scores. Therefore, even if what Reviewer 1 claimed was obvious, it would not really prove the point by itself. In addition to this, Reviewer 1 does not provide a one line proof of the statement (or a reference to such proof)  that they asses as "obvious". If this is so obvious, we invite Reviewer 1 to mathematically prove what we proved and then objectively assess how easy and obvious this will be. Otherwise, we consider Reviewer's 1 conclusion to be their subjective opinion unsupported by facts.
>
> Therefore, we maintain that point (2) raised by Reviewer 1 is based on a collection of false claims and opinions unsupported by facts. As such, point (2) raised by Reviewer 1 is not a valid reason for reject.

---

> ### Author Response · Authors · 2019-11-15
> **Response to Reviewer 1, point (1)**
>
> We would like to sincerely thank Reviewer 1 for providing an in-depth review of the manuscript. We disagree with the Reviewer's assessment of our manuscript and its contributions. We believe that the overly one-sided negative nature of the review may have resulted from a combination of a number of factors, which we will attempt to uncover and analyze in our response below.
>
> First, the motivation behind our work is to show that the image-level supervision is a simple yet extremely effective tool for learning high quality embeddings in the context when image and text pairs are available. We made a very tangible step to prove this concept based on two very well established datasets. We used the generalized zero-shot evaluation framework to prove the concept simply because it seems to be the prevalent evaluation methodology for assessing the embedding quality and transferability at this time. We never had an intent to compete against the state-of-the-art in the generalized zero shot domain.
>
> The practicality of a dataset that is used for evaluation is an important issue and we are happy that Reviewer 1 pointed it out. In our view, this is a valid concern in any scientific study, and perhaps even more so in AI/ML field. For example, let us consider the AWA dataset where classes are represented with predefined attributes. Here are a few examples of the attributes: white (yes/no), black (eys/no), stripes (yes/no), eats fish (yes/no), etc., etc. OK, that approach seems to work for 50 animal classes and about 100 atributes per class in AWA. What is the currently known number of animal species on Earth? It looks like the latest estimate is around 9 million [https://www.bbc.com/news/science-environment-14616161 ] and 10,000 more are opened each year. How many attributes are required to describe 9 million species? Likely, more than a 100. Is AWA dataset representative of the task of classifying animals with its 50 classes and around 100 attributes?  Is the approach of manually labelling attributes per class  for 9 million classes is going to scale and reduce the annotation effort? Our opinion is that the most likely answer to both is no. Same concern applies equally to SUN and ImageNet. This is especially in contrast to the fact that the datasets currently available on the web (pinterest, flicker, etc.) provide paired instances of images and sentences in millions and dozens of millions. People spend a lot of their time providing written sentence descriptions to images in different contexts and they actually enjoy it, so the databases of this nature will naturally grow in size and number with time without someone running a study to collect the data. The approach presented in this paper naturally fits the task of leveraging such data. Therefore, we argue that the task that we considered based on image level sentence description FLOWERS and CUB datasets is at least not less practical than the tasks represented by AWA, SUN or ImageNet datasets. We also argue that the classical attribute-based zero-shot setup is not scalable in the long run. The use of small datasets with dozens of classes and hundreds of attributes does not represent the true complexity of the zero-shot task and creates a false impression that the zero-shot problem provides a labelling complexity short-cut in the attribute based formulation, whereas in any realistic scenario with millions of classes and thousands of attributes to label it will not. Therefore, we maintain that point (1) raised by the reviewer is not a valid reason for reject.

---

### Decision · Program_Chairs · 2019-12-19

**Decision:**

Reject

**Comment:**

This paper demonstrates that per-image semantic supervision, as opposed to class-only supervision, can benefit zero-shot learning performance in certain contexts.  Evaluations are conducted using CUB and FLOWERS fine-grained zero-shot data sets.  In terms of evaluation, the paper received mixed final scores (two reject, one accept).

During the rebuttal period, both reject reviewers considered the author responses, but in the end did not find the counterarguments sufficiently convincing.  For example, one reviewer maintained that in its present form, the paper appeared too shallow without additional experiments and analyses to justify the suitability of the contrastive loss used for obtaining embeddings applied to zero-shot learning.  Another continued to believe post-rebuttal that reference Reed et al., (2016) undercut the novelty of the proposed approach.

And consistent with these sentiments, even the reviewer who voted for acceptance alluded to the limited novelty of the proposed approach; however, the author response merely states that a future revision will clarify the novelty.  But this then requires another round of reviewing to determine whether the contribution is sufficiently new, especially given that all reviewers raised this criticism in one way or another.  Furthermore, the rebuttal also mentions the inclusion of some additional experiments, but again, we don't know how these will turn out.

Based on these considerations then, the AC did not see sufficient justification for accepting a paper with aggregate scores that are otherwise well below the norm for successful ICLR submissions.